# Deep Gaussian Processes for Functional Maps

## Abstract

Learning mappings between functional spaces, also known as function-on-function regression, plays a crucial role in functional data analysis and has broad applications, *e.g.*, spatiotemporal forecasting, curve prediction, and climate modeling. Existing approaches, such as functional linear models and neural operators, either fall short of capturing complex nonlinearities or lack reliable uncertainty quantification under noisy, sparse, and irregularly sampled data. To address these issues, we propose Deep Gaussian Processes for Functional Maps (DGPFM). Our method designs a sequence of GP-based linear and nonlinear transformations, leveraging integral transforms of kernels, GP interpolation, and nonlinear activations sampled from GPs. A key insight simplifies implementation: under fixed locations, discrete approximations of kernel integral transforms collapse into direct functional integral transforms, enabling flexible incorporation of various integral transform designs. To achieve scalable probabilistic inference, we use inducing points and whitening transformations to develop a variational learning algorithm. Empirical results on real-world and PDE benchmark datasets demonstrate that the advantage of DGPFM in both predictive performance and uncertainty calibration.

## 1 Introduction

Function-on-function regression (Ramsay & Dalzell, 1991; Morris, 2015) extends standard regression into functional spaces, where both input and output variables are functions — objects that are infinite-dimensional in nature. It serves as a fundamental tool in functional data analysis (Ramsay & Silverman, 2002) and has found widespread applications such as temporal, spatiotemporal, and curve prediction in econometrics (Rust, 2022), brain imaging (Wang et al., 2014; Wang, 2013), energy and utility consumption forecasting (Fumo & Biswas, 2015), and weather and climate modeling (Holmstrom et al., 2016; Masselot et al., 2018).

Despite the success of existing methods, most of them have focused on functional linear regression (Yao et al., 2005; Manrique, 2016), which predicts the output function by integrating the input function against a (parameterized) regression function. This effectively performs a linear transformation in an infinite-dimensional space. While intuitive and elegant, simple linear transformations often fall short in capturing the complex nonlinear relationships arising in real-world applications. In scientific machine learning, neural network models — often referred to as neural operators (Li et al., 2020; Azizzadenesheli et al., 2024) — have been proposed to learn the operators of partial differential equations (PDEs), which can be viewed as a special case of function-on-function regression. Although successful, these methods typically rely on high-quality, noise-free, regularly sampled data generated from numerical simulations. Moreover, they focus primarily on point estimation and lack mechanisms for uncertainty quantification. In practical settings, however, data are often noisy, sparse, and irregularly sampled, necessitating models that are not only highly expressive but also robust and capable of providing well-calibrated uncertainty estimates.

To overcome these limitations, we propose DGPFM, a deep Gaussian process model for functional maps. DGPFM is flexible enough to capture complex, highly nonlinear relationships and, importantly, enables principled probabilistic inference, offering reliable uncertainty quantification. Specifically, we model the input function as a Gaussian process (GP) and represent the mapping from inputs to outputs as a sequence of GP-based linear and nonlinear transformations in functional space. The linear transformation is implemented via an integral transform of the kernel to obtain the cross-covariance, followed by GP interpolation, while the nonlinear transformation is realized through a GP activation function. We further identify a key insight of our design: under fixed locations, any

discrete approximation of the kernel integral transform causes the intermediate covariance and cross-covariance matrices to cancel out during GP interpolation. This leaves the integral transform being applied directly to the discretized functions, eliminating the need to track complex kernel structures across layers and greatly simplifying model implementation. As a result, our framework can flexibly incorporate arbitrary discrete integral transform designs without requiring explicit computation of the resulting covariance functions. In particular, we propose a simple dimension-wise integral transform, either based on 1D quadrature rules or through the use of the convolution theorem and Fourier transformations inspired by the neural operator literature. Finally, to enable efficient training and probabilistic inference, we introduce a set of inducing points for each GP activation function and use a whitening transformation to construct a variational posterior of the inducing variables. We then develop a stochastic variational inference algorithm for scalable training.

For evaluation, we assess DGPFM on three numerical simulation datasets and three real-world applications. DGPFM nearly always achieves the best prediction accuracy in terms of normalized root mean square error (NRMSE). More importantly, it outperforms alternative methods in test log-likelihood scores, demonstrating substantially better uncertainty calibration than Bayesian version of neural operators trained with popular methods such as Stochastic Gradient Langevin Dynamics and Monte Carlo dropout. Visualizations of prediction examples further confirm that DGPFM produces reliable and well-calibrated uncertainty estimates.

## 2 Background

**Functional Linear Regression.** Function-on-function regression aims to estimate the mapping between two functional spaces $\mathcal{F}_1$ and $\mathcal{F}_2$. Given an input function $f(\cdot) \in \mathcal{F}_1$ and output function $u(\cdot) \in \mathcal{F}_2$, functional linear regression (FLR) (Yao et al., 2005) introduces a linear mapping: $u(\mathbf{x}) = \int w(\mathbf{x}, \mathbf{x}') f(\mathbf{x}') d\mathbf{x}'$, where $w(\mathbf{x}, \mathbf{x}')$ is the coefficient function, extending standard linear regression to an infinite dimensional space. To make estimation tractable, FLR typically employs basis function expansions to represent $f$, $u$, and $w$ in finite-dimensional forms, $e.g.$, $u(\mathbf{x}) = \sum_{k=1}^{K} c_k \phi_k(\mathbf{x})$, $f(\mathbf{x}') = \sum_{l=1}^{L} \alpha_l \psi_l(\mathbf{x}')$, and $w(\mathbf{x}, \mathbf{x}') = \sum_{k=1}^{K} \sum_{l=1}^{L} \omega_{kl} \phi_k(\mathbf{x}) \psi_l(\mathbf{x}')$, where $\{\phi_k\}$ and $\{\psi_l\}$ are basis functions, and $\omega_{kl}$ are the coefficients of the regression surface. The model can then be expressed as multi-variate linear regression. Commonly used bases include B-spines, Fourier bases, and others.

**Gaussian Processes (GPs).** Gaussian processes offer a powerful probabilistic framework for function estimation. Let $f : \mathbb{R}^d \to \mathbb{R}$ denote the target function. A GP places a prior over $f$ such that: $f(\cdot) \sim \mathcal{GP}(m(\cdot), \text{cov}(\cdot, \cdot))$, where $m(\cdot)$ is the mean function and $\text{cov}(\cdot, \cdot)$ is the covariance function, often specified as a kernel function $k(\mathbf{x}, \mathbf{x}')$. In practice, $m$ is usually set to zero. Given input locations $\mathbf{X} = [\mathbf{x}_1, \ldots, \mathbf{x}_N]^\top$, the corresponding function values $\mathbf{f} = [f(\mathbf{x}_1), \ldots, f(\mathbf{x}_N)]$, follow a multi-variate Gaussian distribution, $p(\mathbf{f}) = \mathcal{N}(\mathbf{f} | \mathbf{0}, \mathbf{K})$ where $[\mathbf{K}]_{ij} = \text{cov}(f(\mathbf{x}_i), f(\mathbf{x}_j)) = k(\mathbf{x}_i, \mathbf{x}_j)$. This projection is fundamental to GP inference. Suppose $\mathbf{f}$ is known, and we want to predict the function value at a new location $\mathbf{x}$. Since $\mathbf{f}$ and $f(\mathbf{x})$ also follow a multi-variate Gaussian distribution, we immediately obtain a conditional Gaussian as the predictive distribution, $p(f(\mathbf{x})|\mathbf{f}) = \mathcal{N}(f(\mathbf{x})|\mu(\mathbf{x}), \sigma^2(\mathbf{x}))$, where the conditional mean gives an interpolation estimate, $\mu(\mathbf{x}) = \text{cov}(f(\mathbf{x}), \mathbf{f})\mathbf{K}^{-1}\mathbf{f}$, and the conditional variance $\sigma^2(\mathbf{x}) = \text{cov}(f(\mathbf{x}), f(\mathbf{x})) - \text{cov}(f(\mathbf{x}), \mathbf{f})\mathbf{K}^{-1}\text{cov}(\mathbf{f}, f(\mathbf{x}))$ quantifies the prediction uncertainty, $\text{cov}(f(\mathbf{x}), \mathbf{f}) = k(\mathbf{x}, \mathbf{X}) = [k(\mathbf{x}, \mathbf{x}_1), \ldots, k(\mathbf{x}, \mathbf{x}_N)]$ and $\mathbf{X} = [\mathbf{x}_1, \ldots, \mathbf{x}_N]^\top$.

## 3 Model

We now introduce DGPFM, our deep GP model for learning mappings between functions. Given an input function $f(\cdot)$ and an output function $u(\cdot)$, we model $f$ as a GP, and construct the mapping from $f$ to $u$ through a sequence of intermediate conditional GP layers, which realize successive linear and nonlinear transformations in functional space.

### 3.1 GP-based Linear and Nonlinear Transformation

Specifically, Let $C$ denote the number of GPs in each layer. At a layer $l$, denote each GP $i$ ($1 \le i \le C$) as $h_{l,i}(\cdot)$ and the associated kernel/covariance function as $\kappa_{l,i}(\cdot, \cdot)$. Note that these GPs are defined conditionally on the latent functions of the preceding layer; when marginalized over previous layers, they do not necessarily remain GPs. To perform a linear transformation, we introduce a coefficient

function $w_l(\cdot, \cdot)$, and model

$$h_{l+1,i}(\mathbf{x}) = \int w_l(\mathbf{x}, \mathbf{x}') h_{l,i}(\mathbf{x}') \mathrm{d}\mathbf{x}', \tag{1}$$

which is similar to FLR. Since $h_{l,i}(\cdot)$ is a GP, its integral transform $h_{l+1,i}(\cdot)$ is also a GP; the covariance and cross-covariance functions[1] are given by

$$\kappa_{l+1,i}(\mathbf{x}, \mathbf{x}') = \mathrm{cov}\left(h_{l+1,i}(\mathbf{x}), h_{l+1,i}(\mathbf{x}')\right) = \iint w_l(\mathbf{x}, \mathbf{z}) \kappa_{l,i}(\mathbf{z}, \mathbf{z}') w_l(\mathbf{z}', \mathbf{x}') \mathrm{d}\mathbf{z}\mathrm{d}\mathbf{z}', \tag{2}$$

$$c_{l,i}(\mathbf{x}, \mathbf{x}') = \mathrm{cov}\left(h_{l+1,i}(\mathbf{x}), h_{l,i}(\mathbf{x}')\right) = \int w_l(\mathbf{x}, \mathbf{z}) \kappa_{l,i}(\mathbf{z}, \mathbf{x}') \mathrm{d}\mathbf{z}. \tag{3}$$

To perform nonlinear transform, we instead model $h_{l+1,i}$ as a nonlinear activation function of the value of $h_{l,i}$, and the activation function is sampled from a GP:

$$h_{l+1,i}(\mathbf{x}) = a_l\left(h_{l,i}(\mathbf{x})\right), \quad a_l \sim \mathcal{GP}\left(0, \vartheta_l(z, z')\right), \tag{4}$$

where $\vartheta_l(\cdot, \cdot)$ is the covariance function of $a_l$. The covariance of $h_{l+1,i}$ conditioned on $h_l(\cdot)$ is

$$\kappa_{l+1,i}(\mathbf{x}, \mathbf{x}') = \vartheta_l(h_{l,i}(\mathbf{x}), h_{l,i}(\mathbf{x}')). \tag{5}$$

## 3.2 Model Framework

In general, we assume the input function $f : \Omega \to \mathbb{R}^{d_0}$ is observed (sampled) at a set of locations $\mathbf{X}_{\mathrm{in}} = \{\mathbf{x}_{\mathrm{in},j}\}_{j=1}^{N_{\mathrm{in}}}$ and the output function $u : \Omega \to \mathbb{R}^{d_1}$ sampled at $\mathbf{X}_{\mathrm{out}} = \{\mathbf{x}_{\mathrm{out},j}\}_{j=1}^{N_{\mathrm{out}}}$. Notably, $\mathbf{X}_{\mathrm{in}}$ and $\mathbf{X}_{\mathrm{out}}$ can be *different* or even *non-overlapping*. These locations can be sparse and irregular, and may vary across different input-output function pairs during both training and testing.

To flexibly accommodate varying sampling locations, and to enable tractable computation of the GP layers, we introduce a set of fixed locations $\mathbf{X}_Q$ to serve as the projection points for each GP layer. We first assign a GP prior over each component of $f$: $f^j \sim \mathcal{GP}(0, \nu_j(\cdot, \cdot))$ where $f^j$ is $j$-th component of $f$ with covariance function $\nu_j$. Let $\mathbf{f}^j$, $\mathbf{f}_Q^j$, and $\widehat{\mathbf{f}}^j$ denote the values of $f_j$ at $\mathbf{X}_{\mathrm{in}}$, $\mathbf{X}_Q$, and the noisy observations at $\mathbf{X}_{\mathrm{in}}$, respectively. Define $\widehat{\mathbf{F}} = [\widehat{\mathbf{f}}^1, \dots, \widehat{\mathbf{f}}^{d_0}]$, $\mathbf{F} = [\mathbf{f}^1, \dots, \mathbf{f}^{d_0}]$, and $\mathbf{F}_Q = [\mathbf{f}_Q^1, \dots, \mathbf{f}_Q^{d_0}]$. Their joint distribution factorizes as $p(\widehat{\mathbf{F}}, \mathbf{F}, \mathbf{F}_Q) = p(\mathbf{F})p(\widehat{\mathbf{F}}|\mathbf{F})p(\mathbf{F}_Q|\mathbf{F}) = \prod_{j=1}^{d_0} \mathcal{N}(\mathbf{f}^j|\mathbf{0}, \nu_j(\mathbf{X}_{\mathrm{in}}, \mathbf{X}_{\mathrm{in}})) \mathcal{N}(\widehat{\mathbf{f}}^j|\mathbf{f}^j, \sigma_j^2\mathbf{I})p(\mathbf{f}_Q^j|\mathbf{f}_j)$ where $\sigma_j^2$ is the noise variance, and $p(\mathbf{f}_Q^j|\mathbf{f}_j)$ is conditional Gaussian. Marginalizing out $\mathbf{F}$ yields

$$p(\widehat{\mathbf{F}}, \mathbf{F}_Q) = p(\widehat{\mathbf{F}})p(\mathbf{F}_Q|\widehat{\mathbf{F}}) = \prod_j \mathcal{N}\left(\widehat{\mathbf{f}}_j|\mathbf{0}, \nu_j(\mathbf{X}_{\mathrm{in}}, \mathbf{X}_{\mathrm{in}}) + \sigma_j^2\mathbf{I}\right) \mathcal{N}(\mathbf{f}_Q^j|\mathbf{m}_j, \mathbf{S}_j), \tag{6}$$

where $\mathbf{m}_j = \nu_j(\mathbf{X}_Q, \mathbf{X}_{\mathrm{in}})\mathbf{K}_j^{-1}\widehat{\mathbf{f}}_j$, $\mathbf{S}_j = \nu_j(\mathbf{X}_Q, \mathbf{X}_Q) - \nu_j(\mathbf{X}_Q, \mathbf{X}_{\mathrm{in}})\mathbf{K}_j^{-1}\nu_j(\mathbf{X}_{\mathrm{in}}, \mathbf{X}_Q)$, and $\mathbf{K}_j = \nu_j(\mathbf{X}_{\mathrm{in}}, \mathbf{X}_{\mathrm{in}}) + \sigma_j^2$. To enrich representation, we then introduce a weight matrix $\mathbf{W}_0 \in \mathbb{R}^{d_0 \times C}$ to mix the $d_0$ input channels: $\mathbf{H}_1 = \mathbf{F}_Q\mathbf{W}_0 \in \mathbb{R}^{Q \times C}$, where each of the $C$ columns correspond to a GP projection formed as a linear combination of the $d_0$ components of $f(\cdot)$, evaluated at $\mathbf{X}_Q$.

Next, we apply a sequence of linear and nonlinear transformation as described in Section 3.1. To simplify training and avoid the costly, complex computation of conditional covariance matrices, we model the linear transformation using the GP conditional mean (interpolation) rather than the full conditional Gaussian distribution. Specifically, $p(\mathbf{H}_{l+1}|\mathbf{H}_l) = \prod_{i=1}^C p(\mathbf{h}_{l+1,i}|\mathbf{h}_{l,i})$, and

$$p(\mathbf{h}_{l+1,i}|\mathbf{h}_{l,i}) = \delta\left(\mathbf{h}_{l+1,i} - c_{l,i}(\mathbf{X}_Q, \mathbf{X}_Q)\kappa_{l,i}(\mathbf{X}_Q, \mathbf{X}_Q)^{-1}\mathbf{h}_{l,i}\right) \tag{7}$$

where $\mathbf{h}_{l+1,i}$ and $\mathbf{h}_{l,i}$ denote the $i$-th columns of $\mathbf{H}^{l+1}$, and $\mathbf{H}^l$, respectively, representing the latent functions $h_{l+1,i}(\cdot)$ and $h_{l,i}(\cdot)$'s projections at $\mathbf{X}_Q$, and $\delta(\cdot)$ denotes the Dirac-delta prior. To perform nonlinear transformation, for each GP activation function $a_l(\cdot)$ in (4), we introduce inducing locations $\boldsymbol{\beta} = [\beta_1, \dots, \beta_S]^\top \in \mathbb{R}$, with corresponding inducing values $\boldsymbol{\eta}_l = [a_l(\beta_1), \dots, a_l(\beta_S)]^\top$. These follow the prior: $p(\boldsymbol{\eta}_l) = \mathcal{N}(\boldsymbol{\eta}_l \mid \mathbf{0}, \vartheta_l(\boldsymbol{\beta}, \boldsymbol{\beta}))$. We then model the nonlinear transformation as

$$p(\mathbf{H}_{l+1}|\mathbf{H}_l, \boldsymbol{\eta}_l) = \mathcal{N}\left(\mathrm{vec}(\mathbf{H}_{l+1})|\vartheta_l\left(\mathrm{vec}(\mathbf{H}_l), \boldsymbol{\beta}\right)\vartheta_l(\boldsymbol{\beta}, \boldsymbol{\beta})^{-1}\boldsymbol{\eta}_l, \mathbf{T}_l\right), \tag{8}$$

---

[1] Please see Appendix section A for the detailed derivation.

where $\mathbf{T}_l = \vartheta_l(\mathrm{vec}(\mathbf{H}_l), \mathrm{vec}(\mathbf{H}_l)) - \vartheta_l(\mathrm{vec}(\mathbf{H}_l), \boldsymbol{\beta})\vartheta_l(\boldsymbol{\beta}, \boldsymbol{\beta})^{-1}\vartheta_l(\boldsymbol{\beta}, \mathrm{vec}(\mathbf{H}_l))$, and $\mathrm{vec}(\cdot)$ denotes vectorization.

The final layer $\mathbf{H}_L$ is obtained via linear transform as in (7), but projected onto the observed locations $\mathbf{X}_{\mathrm{out}}$. We then apply a weight matrix $\mathbf{W}_1 \in \mathbb{R}^{C \times d_1}$ to aggregate the $C$ latent channels into the output space: $\mathbf{U} = \mathbf{H}_L\mathbf{W}_1 \in \mathbb{R}^{N_{\mathrm{out}} \times d_1}$. Let $\mathbf{Y}$ denote the observed outputs. We adopt a Gaussian likelihood, $p(\mathbf{Y}|\mathbf{U}) = \prod_{i=1}^{d_1} \mathcal{N}(\mathbf{y}_i|\mathbf{u}_i, v_i\mathbf{I})$, where $\mathbf{y}_i$ is the $i$-th component of the output function observed at the $N_{\mathrm{out}}$ locations, $\mathbf{u}_i$ is the $i$-th column of $\mathbf{U}$, and $v_i$ is the corresponding noise variance. In Appendix Section E, we present the conditional GP priors defined by our model in function space.

## 4 Algorithm

### 4.1 Discrete Approximation of Integral Transforms

A critical challenge in implementing our model lies in computing and tracking the covariance function $\kappa_{l,i}(\cdot, \cdot)$ for each GP layer ($l = 1, 2, \ldots$) and the cross-covariance function $c_{l,i}(\cdot, \cdot)$ required during linear transformations. These computations involve repeated integral transforms (see (2) and (3)) and nested compositions (see (5)), making closed-form derivations intractable and computation highly inefficient. To address this challenge, we adopt a discrete approximation of the integral transform for computing the cross-covariance (see (3)). This leads to a striking insight: the covariance and cross-covariance matrices cancel out, dramatically simplifying the implementation.

Specifically, we consider a quadrature rule to approximate the integral transform. Let $\boldsymbol{\alpha} = [\alpha_1, \ldots, \alpha_M]^\top$ denote the quadrature weights and $\overline{\mathbf{X}} = \{\overline{\mathbf{x}}_j\}_{j=1}^M$ the corresponding nodes. The cross-covariance function in (3) is then approximated as

$$c_{l,i}(\mathbf{x}, \mathbf{x}') \approx \sum_{m=1}^M \alpha_m \cdot w_l(\mathbf{x}, \overline{\mathbf{x}}_m)\kappa_{l,i}(\overline{\mathbf{x}}_m, \mathbf{x}'). \tag{9}$$

According to (7), the values of $h_{l+1,i}(\cdot)$ at the projection points $\mathbf{X}_Q$ are given by

$$\mathbf{h}_{l+1,i} = c_{l,i}(\mathbf{X}_Q, \mathbf{X}_Q)\kappa_{l,i}(\mathbf{X}_Q, \mathbf{X}_Q)^{-1}\mathbf{h}_{l,i}. \tag{10}$$

Substituting (9) into (10), we obtain: $\mathbf{h}_{l+1,i} = \mathbf{W}_l \cdot \mathrm{diag}(\boldsymbol{\alpha}) \cdot \kappa_{l,i}(\overline{\mathbf{X}}, \mathbf{X}_Q)\kappa_{l,i}(\mathbf{X}_Q, \mathbf{X}_Q)^{-1}\mathbf{h}_{l,i}$, where $\mathbf{W}_l$ denotes the evaluations of the weight function $w_l(\cdot, \cdot)$ over the Cartesian product $\mathbf{X}_Q \times \overline{\mathbf{X}}$. Now, if we set the projection points $\mathbf{X}_Q = \overline{\mathbf{X}}$, the cross-covariance and covariance matrices cancel out, yielding:

$$\mathbf{h}_{l+1,i} = \mathbf{W}_l \cdot \mathrm{diag}(\boldsymbol{\alpha}) \cdot \mathbf{h}_{l,i}, \tag{11}$$

which is a discrete approximation applied directly to the integral transform of $h_{l,i}(\cdot)$ as defined in (1).

This observation holds for any discrete approximation for the cross-covariance function (3): as long as we set the projection points $\mathbf{X}_Q$ to the locations used for the approximation, the GP interpolation in (7) reduces to the form of (11). That means, we never need to explicitly compute or track the covariance function at intermediate GP layers. The only required covariance functions are: $\nu_j(\cdot, \cdot)$ for the input function $f$ (see (6)) and $\vartheta_l$ for constructing each GP activation (see (4)). This insight not only greatly streamlines the model implementation, but also enables flexible choices of integral transform approximation.

We proposed two approximation methods. The first follows the quadrature-based approach described in (9) and (11). Rather than specifying a parametric form for the weight function $w_l(\mathbf{x}, \mathbf{x}')$, we directly estimate $\mathbf{W}_l$—the function values over the grid $\mathbf{X}_Q \times \mathbf{X}_Q$. However, since the input to $w_l$ is twice the dimensionality of the function $h_{l,i}(\mathbf{x})$, the size of $\mathbf{W}_l$ grows exponentially with input dimension. This leads to high computational and memory costs. To address this issue, we propose a dimension-wise integral transform. For illustration, consider $\mathbf{x} = [x_1, x_2] \in \mathbb{R}^2$. We introduce two separate weight functions, $w_l^1(x_1, x_1')$ and $w_l^2(x_2, x_2')$, and design the transformation as:

$$h_{l+1,i}(x_1, x_2) = \int w_l^1(x_1, x_1')h_{l,i}(x_1', x_2)\mathrm{d}x_1' + \int w_l^2(x_2, x_2')h_{l,i}(x_1, x_2')\mathrm{d}x_2'. \tag{12}$$

We approximate each integral in (12) using 1D quadrature rules. As a result, we only need to estimate two 2D matrices, $\mathbf{W}_l^1$ and $\mathbf{W}_l^2$, which represent the values of $w_l^1(\cdot, \cdot)$ and $w_l^2(\cdot, \cdot)$ over the Cartesian

products of the respective quadrature nodes. This method generalizes naturally to higher-dimensional inputs and scales linearly with the input dimension, enabling substantial efficiency gains.

For the second approximation, we leverage the idea of Fourier Neural Operators (Li et al., 2020; Tran et al., 2021). We assume each weight function is stationary, i.e., $w_l^j(x_j, x_j') = w_l^j(x_j - x_j')$, and set the projection points $\mathbf{X}_Q$ on a regular grid in each dimension. Using the convolution theorem (Bracewell & Kahn, 1966), each integration in R.H.S of (12) can be computed as:

$$\int w_l^1(x_1, x_1') h_{l,i}(x_1', x_2) \mathrm{d}x_1' = \int w_l^1(x_1 - x_1') h_{l,i}(x_1', x_2) \mathrm{d}x_1' = \mathcal{F}^{-1}\left[\mathcal{F}[w_l^1(\cdot)] \cdot \mathcal{F}[h_{l,i}(\cdot, x_2)]\right],$$

$$\int w_l^2(x_2, x_2') h_{l,i}(x_1, x_2') \mathrm{d}x_2' = \mathcal{F}^{-1}\left[\mathcal{F}[w_l^2(\cdot)] \cdot \mathcal{F}[h_{l,i}(x_1, \cdot)]\right], \tag{13}$$

where $\mathcal{F}[\cdot]$ and $\mathcal{F}^{-1}[\cdot]$ denote the Fourier and inverse Fourier transforms, respectively. Similar to (12), this formulation generalizes directly to higher-dimensional inputs. We approximate (13) by applying the discrete Fourier transform (DFT) to $\mathbf{h}_{l,i}$ on $\mathbf{X}_Q$ (along each dimension), multiplying the result with the discretized spectrum of $w_l^1, w_l^2, \ldots$, and then applying the inverse DFT to obtain $\mathbf{h}_{l+1,i}$.

While this dimension-wise approximate integral transform may bring up additional errors relative to (1), it substantially reduces the number of mode parameters and alleviates the risk of overfitting in practice. In Appendix Section D, we provide a mathematical analysis of the discrete approximation error, and ablation studies in Appendix Section F further confirm the advantage of our approach.

## 4.2 Stochastic Variational Learning

To enable uncertainty quantification via probabilistic inference, we develop an efficient variational learning algorithm (Wainwright et al., 2008; Hensman et al., 2013). Let $\widehat{\mathbf{F}}$ and $\mathbf{Y}$ denote an observed pair of input and output functions. As described in Section 3.2, the joint probability of our model is

$$p(\text{joint}) = p(\widehat{\mathbf{F}})p(\mathbf{F}_Q|\widehat{\mathbf{F}}) \prod_{l+1 \in \Gamma_{\text{lin}}} p(\mathbf{H}_{l+1}|\mathbf{H}_l)p(\mathcal{W}_l) \prod_{l+1 \in \Gamma_{\text{non}}} p(\boldsymbol{\eta}_l)p(\mathbf{H}_{l+1}|\mathbf{H}_l, \boldsymbol{\eta}_l) \cdot p(\mathbf{Y}|\mathbf{U}),$$

where $\Gamma_{\text{lin}}$ and $\Gamma_{\text{non}}$ denote the set of linear and nonlinear layers, respectively, and $p(\mathcal{W}_l)$ is the prior over the weight function values, which can also be specified as GPs. For clarity of exposition, we present the method using a single training instance, though the extension to multiple instances is straightforward. Given the training dataset $\mathcal{D}$, our goal is to infer the posterior distribution of the inducing variables $\{\boldsymbol{\eta}_l\}$ together with the point estimates of the kernel parameters and weight function values $\mathcal{W}_l$, and noise variances. Since each $\mathbf{H}_l$ is determined by these parameters, $\boldsymbol{\eta}_l$, and the proceeding layer $\mathbf{H}_{l-1}$ (see (7) and (8)), posterior inference reduces to sampling $\{\boldsymbol{\eta}_l\}$. At prediction, we draw posterior samples of each $\boldsymbol{\eta}_l$, propagate them through the model, and obtain predictive samples $\mathcal{U}$, thereby enabling uncertainty quantification.

However, direct inference over $\boldsymbol{\eta}_l$ faces two challenges. First, $\boldsymbol{\eta}_l$ and the kernel parameters are tightly coupled, making joint optimization inefficient and prone to poor local optima (Long et al., 2022). Second, computing $p(\mathbf{H}_{l+1}|\mathbf{H}_l, \boldsymbol{\eta}_l)$ in (8) requires evaluating a covariance matrix at all the latent function values $\mathbf{H}_l$ (size $N_Q C$), which is computationally expensive. To address these issues, we first adopt a whitening transformation (Murray & Adams, 2010):

$$\boldsymbol{\eta}_l = \mathbf{A}_l \overline{\boldsymbol{\eta}}_l, \quad p(\overline{\boldsymbol{\eta}}_l) = \mathcal{N}(\mathbf{0}, \mathbf{I}), \quad \mathbf{A}_l \mathbf{A}_l^\top = \vartheta_l(\boldsymbol{\beta}, \boldsymbol{\beta}), \tag{14}$$

where $\mathbf{A}_l$ is the Cholesky factor of the covariance matrix. Note the standard Gaussian prior over $\overline{\boldsymbol{\eta}}_l$ follows from $p(\boldsymbol{\eta}_l) = \mathcal{N}(\mathbf{0}, \vartheta_l(\boldsymbol{\beta}, \boldsymbol{\beta}))$. We then construct a variational posterior approximation:

$$p(\{\overline{\boldsymbol{\eta}}_l\}|\mathcal{D}) \approx q(\{\overline{\boldsymbol{\eta}}_l\}) \propto \prod_{l+1 \in \mathcal{S}_{\text{non}}} q(\overline{\boldsymbol{\eta}}_l)p(\mathbf{H}_{l+1}|\mathbf{H}_l, \mathbf{A}_l \overline{\boldsymbol{\eta}}_l). \tag{15}$$

Following the variational inference framework, the evidence lower bound (ELBO), $\mathcal{L} = \mathbb{E}_q\left[\frac{p(\text{Joint})}{q(\{\overline{\boldsymbol{\eta}}_l\})}\right]$, becomes

$$\mathcal{L} = \mathbb{E}_q\left[\frac{p(\widehat{\mathbf{F}})p(\mathbf{F}_Q|\widehat{\mathbf{F}}) \prod_{l+1 \in \mathcal{S}_{\text{lin}}} p(\mathbf{H}_{l+1}|\mathbf{H}_l)p(\mathcal{W}_l) \prod_{l+1 \in \mathcal{S}_{\text{non}}} p(\overline{\boldsymbol{\eta}}_l)\overline{p(\mathbf{H}_{l+1}|\mathbf{H}_l, \mathbf{A}_l \boldsymbol{\eta}_l)} \cdot p(\mathbf{Y}|\mathbf{U})}{\prod_{l+1 \in \mathcal{S}_{\text{non}}} q(\overline{\boldsymbol{\eta}}_l)\overline{p(\mathbf{H}_{l+1}|\mathbf{H}_l, \mathbf{A}_l \overline{\boldsymbol{\eta}}_l)}}\right],$$

where all conditional terms $p(\mathbf{H}_{l+1}|\mathbf{H}_l, \mathbf{A}_l \overline{\boldsymbol{\eta}}_l)$ cancel, substantially simplifying the computation. The ELBO reduces to $\mathcal{L} = \sum_{l \in \mathcal{S}_{\text{non}}} \text{KL}\left(q(\overline{\boldsymbol{\eta}}_l) \| \mathcal{N}(\mathbf{0}, \mathbf{I})\right) + \mathbb{E}_q[\log(\text{other-terms})]$. We use the reparameterization trick (Kingma & Welling, 2013) to compute unbiased estimates of the expectations and their gradient, and optimize the ELBO using stochastic gradient descent.

**Computational Complexity**: The time complexity of our training method is $\mathcal{O}(dN_Q^2 BL)$ for the quadrature-based dimension-wise integral transform (12) and $\mathcal{O}(dN_Q \log N_Q BL)$ for the Fourier transform approach (13) (via FFT). Here, $d$ is the input dimension, $N_Q$ the number of quadrature nodes or sampling collocations, $L$ the number of layers, $S$ the number of inducing points, and $B$ is the mini-batch size for stochastic training. In both cases, the time complexity scales linearly with input dimensionality. The space complexity is $O(L(dN_Q^2 + BN_QC + S^2))$ for the quadrature-based approach, and $O(L(dN_Q + BN_QC + S^2))$ for the Fourier-based method, accounting for the storage of weight function values, the hidden function values $\mathcal{H}_i$, and the variational posterior $q(\{\boldsymbol{\eta}_l\})$.

## 5 Related Work

Functional data analysis (FDA) has been a prominent area of statistical research for several decades, tracing back to foundational works such as Ramsay & Dalzell (1991); Faraway (1997). Key topics in this field include functional regression (Morris, 2015) and functional principal component analysis (FPCA) (Silverman, 1996; Hall & Hosseini-Nasab, 2006). Functional regression refers to a class of regression models in which the predictors and/or the response are functions. A major subfield is function-on-function regression, where both the predictors and the response are functional. Other widely studied formulations include scalar-on-function regression (Goldsmith & Scheipl, 2014; Reiss et al., 2017; Hullait et al., 2021), function-on-scalar regression (Reiss et al., 2010; Bauer et al., 2018), as well as hybrid or mixed cases. A variety of function-on-function regression methods have been developed (Yao et al., 2005; Manrique, 2016; Kim et al., 2018; Luo & Qi, 2019; Beyaztas & Shang, 2020; Aneiros et al., 2022; Wang et al., 2022; Dette & Tang, 2024), with most of them grounded in the functional linear regression framework, an intuitive extension of classical linear regression. These methods primarily differ in their choice of basis functions, normalization, regularization, *etc*.

In scientific machine learning, operator learning has emerged as an vibrant field aimed at learning mappings between function spaces governed by PDEs. These mappings — often referred to as operators — represent relationships involving derivatives and integrals. Many neural architectures have been specifically designed for learning such PDE operators. One of the most prominent classes of operator learning models are the Fourier Neural Operators (FNO) (Li et al., 2020) and their extensions (Tran et al., 2021; Lingsch et al., 2024), which perform functional transformations through Fourier layers combined with standard neural network activations such as GeLU. Other notable operator learning models include the Multiwavelet Neural Operator (Gupta et al., 2021), CNO (Raonic et al., 2023), DeepONet (Lu et al., 2021), and transformer-based approaches (Cao, 2021; Li et al., 2022a; Hao et al., 2023), among others.

The classical Deep GP framework (Damianou & Lawrence, 2013), is designed to learn a single function from its observed values across various input locations, aligning with standard GP regression setting. In contrast, our formulation extends this paradigm by considering both the input and output as functions, aiming to learn their relationship directly in the functional space. Our variational inference approach is similar to (Salimbeni & Deisenroth, 2017), where for each GP activation, we introduce a set of inducing variables to facilitate tractable function estimation. However, we further leverage the whitening transformation, which reparameterizes the prior over the inducing variables as a standard Gaussian distribution. This reparameterization decouples the typically strong correlation between the kernel parameters and inducing variables, thereby easing the optimization of variational ELBO.

## 6 Experiment

We evaluated DGPFM on three numerical PDE simulation scenarios and three real-world applications. The PDE simulation scenarios represent the central focus of neural operator methods. To further demonstrate the versatility of our approach, we extended our evaluation to real-world applications characterized by sparse, noisy, and irregularly sampled data. This comprehensive evaluation examines the robustness and adaptability of our method across both controlled and practical settings.

The simulation scenarios are as follows. (1) **1D Burgers' Equation** (Lu et al., 2022): learning a mapping from the initial condition $u_0(x)$ to the solution at time $t = 1$, $u_1(x)$, where $u_0, u_1 : (0, 1) \rightarrow \mathbb{R}$. (2) **2D Darcy Flow**, a two-dimensional Darcy flow equation (Lu et al., 2022), predicting the pressure field $u : [0, 1]^2 \rightarrow \mathbb{R}$ from the given permeability field $c : [0, 1]^2 \rightarrow \mathbb{R}$. (3) **3D Compressible Navier-Stokes (NS) Equations**. The third scenario involves three-dimensional compressible fluid

dynamics governed by the compressive NS equations (Takamoto et al., 2022). The task is to predict the velocity field $v_1$ at the first time step from the initial velocity field $v_0$, where $v_0, v_1 : [0, 1]^3 \rightarrow \mathbb{R}$. For the 1D Burgers' equation and 2D Darcy flow, we used 1,000 training examples and 200 test examples. For the 3D compressible NS equations, only 100 examples are available; we used 90 for training and 10 for testing . Detailed simulation settings are provided in Appendix Section B.

We also tested on the following real-world applications. (1) **Beijing-Air**[2], including hourly measurements of several air pollutants in Beijing from 2014 to 2017. The task is to predict the hourly concentration of CO over the following week based on the previous week's measurements of SO2, CO, PM2.5, and PM10. This constitutes a 1D function-to-function regression task. We randomly selected 5,000 weeks for training and 1,000 weeks for testing. (2) **SLC-Precipitation**, compiled from daily precipitation records collected by weather stations distributed across the Great Salt Lake area from 1954 to 2023. The prediction task is to infer the next day's precipitation based on the current day's readings. The weather stations are sparsely and irregularly distributed, with substantial missing data. We randomly selected 128 stations, and formulated the task as 2D function-to-function regression. We used 5,000 examples for training and 1,000 for testing. (3) **Quasar Reverberation Mapping**, derived from the Zwicky Transient Facility (ZTF) (Bellm et al., 2019) at the Palomar Observatory. The objective is to model the relationship between a quasar's central continuum emission and the delayed response from surrounding emitting regions, which is known as reverberation mapping (Blandford & McKee, 1982; Peterson, 1993). The dataset consists of 793 pairs of irregularly sampled light curves, with distinct sampling locations for input and output functions. The task is formulated as a 1D function-on-function regression problem. We used 650 examples for training and 143 for testing. More details are provided in Appendix Section B.4.

We compared DGPFM with the following methods: (1) Functional Linear Regression (FLR) (Morris, 2015), the mainstream function-on-function regression method that extends linear regression into functional spaces. We employed two popular basis expansions: one with Fourier bases (denoted as LFR-Fourier) and the other with B-splines (LFR-BSpline). (2) Fourier Neural Operator (FNO) (Li et al., 2022b), the most widely used neural operator model that introduces Fourier layers to perform linear functional transformations. However, FNO requires inputs and output functions to be sampled on a uniform grid due to its reliance on Fast Fourier Transform (FFT). (3) DSE-FNO (Lingsch et al., 2024), a most recent variant of FNO, which uses non-uniform discrete Fourier transforms (NUDFT) to enable direct spectral evaluations on irregular domains. However, DSE-FNO still assumes that both input and output functions share identical sampling locations to main the consistency between NUDFT and its inverse. (4) GNOT (Hao et al., 2023), a transformer-based neural operator capable of flexibly handling arbitrarily irregular sampling in both input and output functions via cross-attention mechanisms (Vaswani et al., 2017). (5) LFR-GP, a baseline model that uses a single integral GP transform layer with Gauss-Legendre quadrature (see (11)), effectively representing a simplified version of DGPFM with one GP layer. We evaluated two versions of our method: DGPFM-QR, which performs dimension-wise integral transforms using numerical quadrature rules (see (12)); DGPFM-FT, which leverages the convolution theorem and Fourier transform to compute the integral transforms (see (13)). For each method on each task, we used a separate validation set to select the optimal hyperparameters based on one training dataset. These hyperparameters were then fixed, and the model was trained and tested across five independent runs. We report both the mean prediction error and the standard deviation to assess performance stability and accuracy. We leave the details about the hyperparameter selection for each method in Appendix Section C.

**Prediction Accuracy**. We first evaluated the normalized root-mean-square error (NRMSE) for each method. As shown in Table 1, our method (DGPFM-FT/-QR) consistently achieves the highest prediction accuracy, with the exception of the NS dataset, where FLR-Fourier/-BSpline outperform all methods. This exception may be due to the NS dataset's difficulty — characterized by a high Mach number — and the limited number of training examples (only 90 available). In such cases, linear models with simple basis functions can offer more robust predictive performance. Nevertheless, DGPFM still outperforms all the other methods by a considerable margin. On *1D-Burgers* and *2D-Darcy* , DGPFM-FT surpasses highly optimized neural operator models, while DGPFM-QR achieves comparable error levels. Notably, across all three real-world applications, both DGPFM-FT and DGPFM-QR significantly outperform all competing methods, with statistical significance exceeding the 95% confidence level. Furthermore, neural operators such as FNO and DSE-FNO

---

[2]`https://archive.ics.uci.edu/dataset/501/beijing+multi+site+air+quality+data`

Table 1: Normalized Root-Mean-Square (NRMSE) error. The top two results are highlighted in bold. N/A indicates that the method is not applicable[3].

(a) Simulation datasets.

| Method | 1D-Burgers | 2D-Darcy | 3D-NS |
|---|---|---|---|
| FLR-Fourier | 3.76e-1 $\pm$ 1.90e-3 | 4.58e-1 $\pm$ 6e-4 | **4.51e-1 $\pm$ 2.22e-2** |
| FLR-BSpline | 4.08e-1 $\pm$ 2.00e-3 | 4.54e-1 $\pm$ 6e-4 | **4.52e-1 $\pm$ 2.26e-2** |
| FLR-GP | 3.36e-1 $\pm$ 1.00e-3 | 5.54e-1 $\pm$ 1.70e-3 | 6.17e-1 $\pm$ 4.61e-2 |
| GNOT | 8.90e-3 $\pm$ 1.40e-4 | 2.58e-2 $\pm$ 3.40e-4 | 8.15e-1 $\pm$ 2.18e-2 |
| FNO | **2.76e-3 $\pm$ 4.17e-5** | 1.79e-2 $\pm$ 2.49e-4 | 5.81e-1 $\pm$ 4.77e-2 |
| DSE-FNO | 6.69e-2 $\pm$ 9.89e-4 | 3.63e-2 $\pm$ 1.70e-4 | N/A |
| DGPFM-FT | **1.79e-3 $\pm$ 1.86e-4** | **1.67e-2 $\pm$ 1.40e-4** | 4.67e-1 $\pm$ 2.02e-3 |
| DGPFM-QR | 7.66e-3 $\pm$ 6.21e-4 | 1.83e-2 $\pm$ 5.17e-4 | 4.85e-1 $\pm$ 1.83e-3 |

(b) Real-world datasets.

| Method | Beijing-Air | Quasar | SLC-Precipitation |
|---|---|---|---|
| FLR-Fourier | 0.636 $\pm$ 3.00e-3 | 0.008 $\pm$ 1.00e-3 | 1.02 $\pm$ 3.40e-3 |
| FLR-BSpline | 0.639 $\pm$ 3.00e-3 | 0.008 $\pm$ 1.00e-4 | 1.02 $\pm$ 3.50e-3 |
| FLR-GP | 0.552 $\pm$ 1.90e-3 | 0.0079 $\pm$ 1.00e-4 | 1.03 $\pm$ 1.06e-3 |
| GNOT | 0.553 $\pm$ 1.80e-3 | 0.0054 $\pm$ 1.00e-4 | 0.836 $\pm$ 6.76e-3 |
| FNO | 0.403 $\pm$ 2.10e-3 | N/A | N/A |
| DSE-FNO | 0.529 $\pm$ 1.34e-3 | N/A | N/A |
| DGPFM-FT | **0.304 $\pm$ 1.16e-3** | **0.0047 $\pm$ 6.06e-5** | **0.776 $\pm$ 4.64e-3** |
| DGPFM-QR | **0.263 $\pm$ 9.13e-3** | **0.0045 $\pm$ 6.20e-5** | **0.777 $\pm$ 4.15e-3** |

Table 2: Mean Negative Log-Likelihood (MNLL). The top two results are highlighted in bold.

(a) Simulation datasets.

| Method | 1D-Burgers | 2D-Darcy | 3D-NS |
|---|---|---|---|
| GNOT-MCDropout | 2.02 $\pm$ 0.031 | 2.84 $\pm$ 0.104 | 4.47e+4 $\pm$ 6.48e+3 |
| FNO-MCDropout | 312.46 $\pm$ 19.32 | 111.11 $\pm$ 4.10 | 102.8 $\pm$ 31.3 |
| GNOT-SGLD | 5.33 $\pm$ 3.50e-3 | 13.29 $\pm$ 0.022 | 578.6 $\pm$ 101.6 |
| FNO-SGLD | 12.05 $\pm$ 2.96 | 13.24 $\pm$ 4.62 | 40.0 $\pm$ 23.18 |
| DGPFM-FT | **-4.27 $\pm$ 0.101** | **-3.50 $\pm$ 0.025** | **9.58 $\pm$ 1.88** |
| DGPFM-QR | **-3.29 $\pm$ 0.066** | **-4.13 $\pm$ 0.042** | **11.2 $\pm$ 2.01** |

(b) Real-world datasets.

| Method | Beijing-Air | Quasar | SLC-Precipitation |
|---|---|---|---|
| GNOT-MCDropout | 464.7 $\pm$ 35.1 | 54.3 $\pm$ 5.97 | 404.5 $\pm$ 16.9 |
| FNO-MCDropout | 171.1 $\pm$ 6.12 | N/A | N/A |
| GNOT-SGLD | 10.38 $\pm$ 1.10e-3 | 17.06 $\pm$ 0.03 | 5298.74 $\pm$ 943.705 |
| FNO-SGLD | 18.66 $\pm$ 3.08 | N/A | N/A |
| DGPFM-FT | **7.78 $\pm$ 5.81e-2** | **-0.534 $\pm$ 6.69e-2** | **17.3 $\pm$ 2.18** |
| DGPFM-QR | **8.01 $\pm$ 9.36e-2** | **-0.924 $\pm$ 1.71e-2** | **25.9 $\pm$ 4.82** |

are limited by regular or fixed sampling locations, making them unsuitable for real-world scenarios involving arbitrary observation points. For instance, neither FNO nor DSE-FNO can be applied to the Quasar and SLC-Precipitation datasets, where the number and locations of sampling points vary not only between the input and output functions but also across different training and testing instances. Together the results highlight the strong flexibility and predictive performance of DGPFM.

**Uncertainty Quantification**. Next, we evaluated our method from a probabilistic perspective by examining the Mean Negative Log-Likelihood (MNLL). For comparison, we trained FNO and GNOT

---

[3]FNO is not applicable to irregularly sampled locations, while DSE-FNO cannot be used when the input and output functions are sampled at different locations, leading to missing results on several datasets. Since the official DSE-FNO implementation does not support 3D functional mappings (required in *3D-NS* ), we developed our own version. However, it produced unreasonably large errors, and thus the corresponding results are omitted.

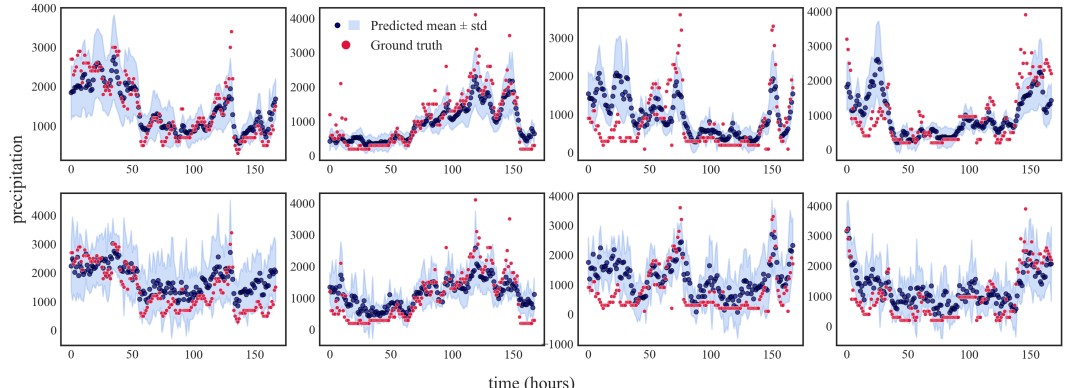

Figure 1: Prediction examples of DGPFM on *Beijing-Air* dataset. The shaded regions indicate one predictive standard deviation. The top row shows the prediction of DGPFM-FT and the bottom row DGPFM-QR.

using two widely adopted Bayesian neural network training approaches: Stochastic Gradient Langevin Dynamics (SGLD) (Welling & Teh, 2011) and Monte Carlo Dropout (MCDropout) (Gal & Ghahramani, 2016). For SGLD, the initial learning rate was chosen from the range $\{10^{-6}, 10^{-5}, \ldots, 10^{-2}\}$, while for MCDropout, the dropout rate was tuned across $\{0.1, 0.2, \ldots, 0.5\}$. As shown in Table 2, both variants of our method —DGPFM-FT and DGPFM-QR— consistently achieve the lowest MNLL across all the datasets, largely outperforming the competing methods. These results highlight the strength of our method not only in predictive accuracy but also in uncertainty quantification.

We further investigated the probabilistic predictions of our method. Specifically, we randomly selected four test examples respectively from two real-world applications: *Beijing-Air* and *Quasar*, as well as two test examples respectively from simulation applications: *2D Darcy* and *1D Burgers*. Figure 1 and Appendix Figure 2 display the predictive means and standard deviations produced by DGPFM-FT and DGPFM-QR at each input location. As observed, in regions where the predicted mean closely matches the ground truth, the shaded region—representing the predictive standard deviation (STD) —is relatively narrow. Conversely, in regions where the discrepancy between the predictive mean and the ground truth is larger, the shaded area expands, indicating higher predictive uncertainty. It is interesting to observe that the predictive standard deviation produced by DGPFM-FT is smoother than that of DGPFM-QR. This may be because DGPFM-QR freely learns the weight function at the quadrature nodes, whereas DGPFM-FT leverages Fourier transforms to retain primarily low-frequency components, resulting in smoother uncertainty estimates. Appendix Figure 3 compares predictive STD with point-wise error and their normalized counterparts. In both cases, larger errors (or relative errors) generally correspond to larger predictive STDs. After normalization, however, regions with large absolute errors often exhibit smaller relative errors and thus smaller normalized STDs. This indicates that predictive STD naturally scales with the magnitude of ground-truth values, which is reasonable. Additional examples are shown in Figure 4 (Appendix), where very small errors (NMSE: 0.005) correspond to very small predictive STDs.

Overall, the standard deviation outputs from our method appropriately reflect the quality of the predictions, confirming that DGPFM provides well-calibrated uncertainty estimates.

**Ablation Studies.** We conducted extensive ablation studies to further evaluate DGPFM, examining various hyperparameter choices, the number of projection points, integral transforms, and the effectiveness of GP activation function. We also analyzed the learned weight functions, and evaluated the training efficiency. Details are provided in Appendix Section F, G, and H.

# 7 Conclusion

We have presented DGPFM, a deep Gaussian process model designed for learning mappings between functions. The model comprises a sequence of GP layers that perform linear transformations and nonlinear activations in functional space. DGPFM is capable of handling sparse, noisy, and arbitrarily irregularly sampled data, while providing probabilistic inference for effective uncertainty quantification. Its performance on both simulated and real-world datasets is encouraging, outperforming or performing on par with classical functional linear regression and recent neural operator methods.

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

# Appendix

## A  GP Covariance for Integral Transformation

Suppose a stochastic function $f$ is sampled from a GP prior with covariance function as a kernel function $\kappa(\cdot, \cdot)$,

$$f \sim \mathcal{GP}(0, \kappa(\mathbf{x}, \mathbf{x}')).$$

From the weight space view (Rasmussen & Williams, 2006), we can represent:

$$f(\mathbf{x}) = \phi(\mathbf{x})^\top \mathbf{w},$$

where $\phi(\mathbf{x})$ is the implicit feature mapping of the kernel, *i.e.*, $\kappa(\mathbf{x}, \mathbf{x}') = \phi(\mathbf{x})^\top \phi(\mathbf{x}')$, and $\mathbf{w} \sim \mathcal{N}(0, \mathbf{I})$. Note that the feature mapping $\phi(\cdot)$ can be infinitely dimensional. Suppose function $h$ is a linear transformation of $f$,

$$h(\mathbf{x}) = \int_\Omega \mathcal{W}(\mathbf{x}, \mathbf{z}) f(\mathbf{z}) \mathrm{d}\mathbf{z} = \int_\Omega \mathcal{W}(\mathbf{x}, \mathbf{z}) \phi(\mathbf{z})^\top \mathbf{w} \mathrm{d}\mathbf{z}, \tag{16}$$

where $\mathcal{W}$ is the coefficient (or weight) function. We can accordingly compute the cross-covariance function between $h$ and $f$ at any pair of inputs $(\mathbf{x}, \mathbf{x}')$ by

$$\mathrm{cov}(h(\mathbf{x}), f(\mathbf{x}')) = \mathbb{E}\left[h(\mathbf{x}) f(\mathbf{x}')\right] - \mathbb{E}[h(\mathbf{x})]\mathbb{E}[f(\mathbf{x}')]$$
$$= \int_\Omega \mathcal{W}(\mathbf{x}, \mathbf{z}) \phi(\mathbf{z})^\top \mathbb{E}\left[\mathbf{w}\mathbf{w}^\top\right] \phi(\mathbf{x}') \mathrm{d}\mathbf{z}$$
$$= \int_\Omega \mathcal{W}(\mathbf{x}, \mathbf{z}) \phi(\mathbf{z})^\top \phi(\mathbf{x}') \mathrm{d}\mathbf{z}$$
$$= \int_\Omega \mathcal{W}(\mathbf{x}, \mathbf{z}) \kappa(\mathbf{z}, \mathbf{x}') \mathrm{d}\mathbf{z}. \tag{17}$$

Similarly, we can derive the covariance function of $h$:

$$\mathrm{cov}(h(\mathbf{x}), h(\mathbf{x}')) = \int_\Omega \int_\Omega \mathcal{W}(\mathbf{x}, \mathbf{z}) \kappa(\mathbf{z}, \mathbf{z}') \mathcal{W}(\mathbf{z}', \mathbf{x}') \mathrm{d}\mathbf{z}\mathrm{d}\mathbf{z}'. \tag{18}$$

From (17) and (18) we can see that these (cross-) covariance functions are non-stationary even if both $\mathcal{W}$ and $\kappa$ are stationary, namely when $\mathcal{W}(\mathbf{x}, \mathbf{z}) = \mathcal{W}(\mathbf{x} - \mathbf{z})$ and $\kappa(\mathbf{z}, \mathbf{x}') = \kappa(\mathbf{z} - \mathbf{x}')$.

## B  Dataset Details

### B.1  1D Burger's equation

We first considered a 1D Burger's equation:

$$u_t + u_{xx} = \nu u_{xx}, u(x, 0) = u_0(x), \tag{19}$$

where $(x, t) \in [0, 1]^2$, and $u_0(x)$ is the initial condition, and $\nu = 0.1$ is the viscosity. We aim to learn a mapping from the initial condition to the solution at $t = 1$, namely, $u_0 \to u_1(x) \triangleq u(x, 1)$. The initial condition $u_0$ is sampled from a Gauss random field, $\mathcal{N}(0, 625(-\Delta + 25\mathbf{I})^{-2})$. The dataset was generated in (Lu et al., 2022), with each pair of input and output functions sampled at the same set of 128 equally spaced locations across the spatial domain.

### B.2  2D Darcy Flow

We then employed a 2D Darcy flow equation in a rectangle domain:

$$-\nabla(c(\mathbf{x})\nabla u(\mathbf{x})) = 1, \tag{20}$$

where $\mathbf{x} \in [0, 1]^2$, $c(\mathbf{x}) > 0$ is the permeability field, and $u(\mathbf{x}) = 0$ at the boundary. The goal is to predict the solution field from the permeability field: $c \to u$. The permeability field $c$ a piecewise constant function derived by first sampling a continuous function from a Gauss random field $\mathcal{N}(0, (-\Delta + 9\mathbf{I})^2)$, and then mapping the positive values to 12 and the negative values to 3. Every input-output function pair is discretized on a $29 \times 29$ uniform grid over the input domain. The dataset was generated and shared by Lu et al. (2022).

### B.3   3D Compressible Naiver-Stoke (NS) Equations

The third scenario involves 3D compressible NS equations:

$$
\begin{aligned}
\partial_t \rho + \nabla \cdot (\rho \mathbf{v}) &= 0, \\
\rho \left( \partial_t \mathbf{v} + \mathbf{v} \cdot \nabla \mathbf{v} \right) &= -\nabla p + \eta \Delta \mathbf{v} + (\zeta + \eta/3) \nabla (\nabla \cdot \mathbf{v}), \\
\partial_t \left[ \epsilon + \frac{\rho v^2}{2} \right] + \nabla \cdot \left[ \left( \epsilon + p + \frac{\rho v^2}{2} \right) \mathbf{v} - \mathbf{v} \cdot \sigma' \right] &= 0,
\end{aligned}
\tag{21}
$$

where $\rho$ is the mass density, $\mathbf{v}$ is the velocity, $p$ is the gas pressure, $\epsilon = p/(\Gamma - 1)$ is the internal energy, $\Gamma = 5/3$, $\sigma'$ is the viscous stress tensor, and $\eta, \zeta$ are the shear and bulk viscosity, respectively. The behavior of the fluid is sensitive to the Mach number $M = |v|/c_s$, where $c_s = \sqrt{\Gamma p/\rho}$. The data were generated and made available through PDEBench (Takamoto et al., 2022), a widely used benchmark dataset for scientific machine learning. We considered the high Mach number case ($M = 1.0$), where the fluid behavior is complex, making the learning task challenging. The input and output functions are discretized on a uniform grid of size $64 \times 64 \times 64$.

### B.4   Quasar Reverberation Mapping

In astronomy, understanding the relationship between the central continuum emission of a quasar and the subsequent response from surrounding emitting regions is key to inferring its physical properties, structure, and kinematics — a process known as reverberation mapping (Blandford & McKee, 1982; Peterson, 1993). This task is often modeled via an unknown transfer function that links the response emission to the driving continuum emission, naturally framing the problem as function-on-function regression.

The Zwicky Transient Facility (ZTF) (Bellm et al., 2019), located at the Palomar Observatory, is an automated time-domain survey utilizing a 4-foot Schmidt telescope equipped with a 47.2-square-degree field-of-view camera. ZTF scans the entire Northern sky with a cadence of approximately three nights during Phase I (May 2018–September 2020) and two nights during Phase II (December 2020–present) for its custom $g$-band and $r$-band photometric filters, with a four-night cadence for the $i$-band.

To construct a dataset aligned with this task, we collected $g$-band and $r$-band light curves from the most recent ZTF data release (DR23) (Masci et al., 2018), focusing on the first 18,000 objects in the Million Quasars catalogue (Flesch, 2023). In this setting, we treat the shorter-wavelength $g$-band light curve as the input function driving the response observed in the $r$-band light curve, which serves as the output function.

We preprocessed the raw data following the methodology of (Sánchez-Sáez et al., 2021). Specifically, we retained light curves with mean magnitudes slightly brighter than the ZTF limiting magnitude of 20.6 and fainter than 13.5 to avoid saturated measurements. We excluded observations with magnitude errors exceeding one and with non-zero catflags quality scores. However, we did not filter light curves based on variability features.

We further restricted the data to observations within the first 2,000 days and randomly sampled up to 500 time points from each light curve, provided sufficient data existed. This resulted in 793 pairs of irregularly sampled light curves, with differing time points across the input and output functions for each example—thereby offering a suitable testbed for function-on-function regression. For our experiments, we randomly split the dataset into 650 training and 143 testing examples.

## C   Hyperparameter Selection

Here we provide hyperparameter selection details for each method.

- **FLR**: We adopted the implementation from the Scikit-FDA library[4] for LFR-Fourier and LFR-BSpline. The primary hyperparameter is the number of bases, which was selected from {2, 3, ..., 30}. The range of each basis function is set to a minimum range that covers the observed output function values, *e.g.*, [0, 1] for *1D Burgers* and *2D Darcy*.

---

[4]https://fda.readthedocs.io/en/latest/

The intercept parameter was jointly estimated with the basis coefficients. We employ the second order differential operator regularization, which is the default choice of the library. The implementation of FLR-GP is directly from that of DGPFM. The selection of the hyperparameters is shared with that for DGPFM, except we fixed the number of GP layers to one, and there is not any hyperparameter tuning for GP activation functions.

- **FNO**[5]: The hyperparameters include the number of modes, which varies from {8, 10, 12, 16, 20}, the number of channels for channel lifting, which varies from {8, 16, 32, 64, 128, 256}, and the number of Fourier layers, which varies from {2, 3, 4}. We used GELU activation, the default choice in the official library.

- **GNOT**[6]: the hyperparameters include the number of attention layers, varying from {3, 4, 5}, the dimensions of the embeddings, varying from {8, 16, 32, 64}, and the inclusion of mixture-of-expert-based gating, specified as either {yes, no}. We used GeLU activation, the default choice of the official library.

- **DSE-FNO**[7]: The set of hyperparameters are the same as FNO, including the number of modes chosen from {8, 10, 12, 16, 20}, the number of latent channels from {8, 16, 32, 64, 128, 256}, and the number of integration layers from {2, 3, 4}. The activation was chosen from {GeLU, ReLU, SiLU}.

- **DGPFM**: Our method was implemented using JAX (Frostig et al., 2018). We used the ADAM optimizer, with the initial learning rate selected from {5e-5, 1e-4, 5e-4}, and a cyclical cosine annealing schedule with the max learning rate as $0.001$. The number of training epochs was chosen from {100, 250, 500, 1000, 5000, 10000}. The number of GP layers was varied from {2, 3, 4}. All the kernel functions for both the input and output GPs, as well as the GP activation, were the Square Exponential (SE) kernel or a weighted combination of two Matérn kernels with degrees of freedom $5/2$ and $13/2$. The number of inducing points for each GP activation was selected from $\{32, 64, 128, 256, 512\}$, and the column dimension of the weight matrices $\mathbf{W}_0$ and $\mathbf{W}_1$ from $\{4, 8, 16, 32, 64, 128, 256\}$. For DGPFM-FT on datasets sampled at regular grids, we kept the number of projection locations the same as the number of locations in the original functions, and on the irregular sampled datasets searched over $\{32, 64, \ldots, 512\}$ (using their tensor product for higher dimensional problems). In a similar capacity, for DGPFM-QR we used a trapezoidal quadrature rule at the function locations when handling problems on a regular grid, and for the irregular grid problems, used a tensor-product Gauss-Legendre rule with the number of one-dimensional nodes as selected from $\{32, 64, \ldots, 512\}$.

All the neural operator methods (FNO, GNOT, DSE-FNO) used ADAM optimization with learning rate selected from $\{10^{-5}, 5 \times 10^{-5}, 10^{-4}, 4 \times 10^{-3}, 10^{-3}\}$. The maximum number of epochs was set to 10000, which ensures convergence. The batch size was set to 500 for *Beijing-Air* dataset and 100 for all the other datasets. We ran all the methods on NCSA Delta GPU cluster[8], with NAVIDA A40 GPUs.

## D  Error Analysis of Discrete Integral Transform

To perform discrete integral transforms, we used several classical numerical approximation methods: Gauss–Legendre (GL) Qudrature, Trapezoidal (TR) Rules, and Fourier-based Convolution. Their theoretical error properties are well established and summarized as follows. Specifically, for a given function $f$,

- **Gauss–Legendre Quadrature**:
    - An $N$-point GL rule is exact for polynomials up to degree $2N - 1$.
    - For $f \in C^k$, error is $O(1/N^k)$ (no periodicity constraint for this like in Trapezoidal rule); for analytic $f$, error is $O(e^{-cN})$.

- **Trapezoidal Rule**:

---

[5]https://github.com/neuraloperator/neuraloperator
[6]https://github.com/HaoZhongkai/GNOT
[7]https://github.com/camlab-ethz/DSE-for-NeuralOperators
[8]https://www.ncsa.illinois.edu/research/project-highlights/delta/

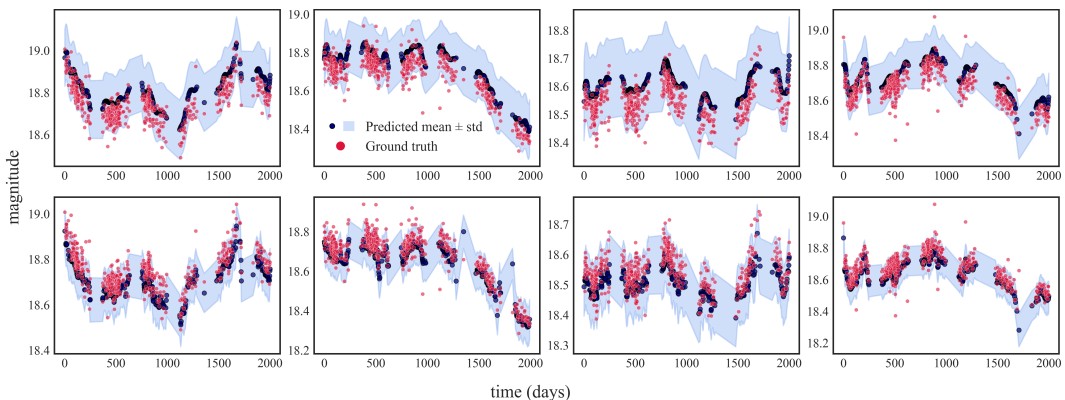

Figure 2: Prediction examples of DGPFM on *Quasar* dataset. The shaded regions indicate one predictive standard deviation. The top row shows the prediction of DGPFM-FT and the bottom row DGPFM-QR.

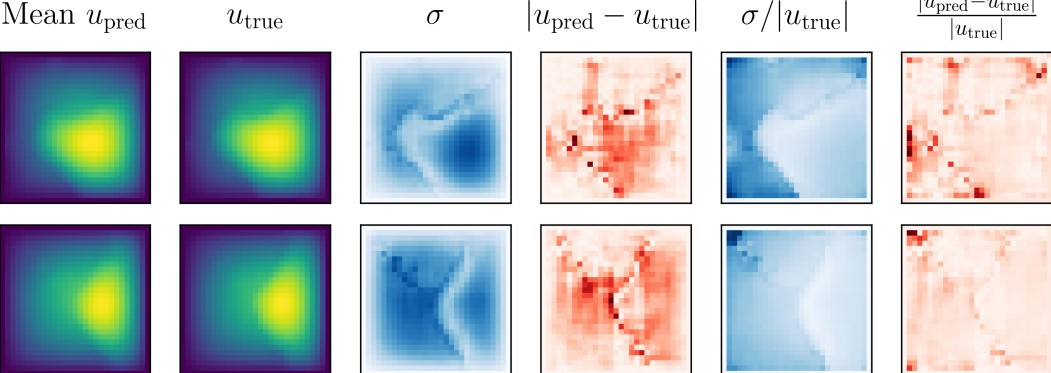

Figure 3: Prediction examples of DGPFM-QR on *2D Darcy*, $\sigma$ denotes the predictive standard deviation (STD). The last two columns show the point-wise predictive std normalized by the ground-truth.

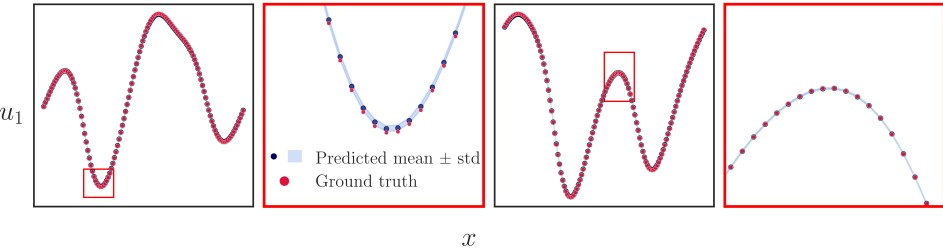

Figure 4: Prediction examples of DGPFM-FT on *1D Burgers*.

- On $[a, b]$, the error is $O(1/N^2)$ if $f''$ exists and is continuous; degrades to $O(1/N)$ if $f \in C^0$.
- For periodic $f \in C^m$, error improves to $O(1/N^m)$; analytic periodic functions yield exponential convergence.

- **Fourier-based Convolution**:
  - If $f$ is periodic and $C^m$, FFT convolution matches TR's periodic rates: $O(1/N^m)$ or exponential if analytic.
  - If $f$ is non-periodic, the implicit periodic extension introduces boundary jumps, causing error stuck at $O(1/N)$ regardless of interior smoothness.

Our model applies a dimension-wise integral transform instead of performing a full-dimensional one. If we consider the full-dimensional transform as the ground truth, the approximation error introduced by our dimension-wise approach can be expressed (in the 2D case) as:

$$\Delta_1 = ((\mathcal{T} - \widetilde{\mathcal{T}})h)(x_1, x_2) = \iint \{w(x_1, x_2, y_1, y_2) - [w_1(x_1, y_1) + w_2(x_2, y_2)]\} h(y_1, y_2)\, dy_1\, dy_2.$$

If the full weight function $w(\cdot)$ can be decomposed as the sum of independent components $w_1$ and $w_2$, the error vanishes, *i.e.,* $\Delta_1 = 0$. Otherwise, a gap remains. However, since $w(\cdot)$, $w_1$, and $w_2$ are unknown a priori and must be learned from data, the approximation error is inherently data-dependent.

From a modeling perspective, our approach adopts a reduced model space (or equivalently, a simpler inductive bias) to represent the weight function(s) in the integral transform, which is *not* necessarily a limitation. Instead, the dimension-wise transform enables the model's parameter complexity to scale linearly with input dimensionality. In contrast, using a full-dimensional transform causes exponential growth in the number of parameters needed to learn the weight function, significantly increasing training cost and the risk of overfitting. Our ablation study has confirmed this point. See Table 7.

The total error regarding our discrete dimension-wise integral transform can be decomposed as two parts. The first one is the aforementioned $\Delta_1$. The second part comes from the numerical approximation error. Each of the dimension-wise integrals (*e.g.*, those involving $w_1$ and $w_2$) is then approximated using numerical quadrature or discrete Fourier transforms, introducing a separate approximation error, denoted as $\Delta_2$. The analysis of $\Delta_2$ has already been provided. The total error can be expressed as $\Delta = |\Delta_1| + |\Delta_2|$.

# E    GP Priors in Functional Space

Our model constructs a sequence of conditional GP prior in the functional space. Specifically, at each layer $l$, When $h_{l,i} \to h_{l+1,i}$ is a nonlinear transformation (see (8)), it implies

$$h_{l+1,i}(\cdot) \mid h_{l,i}(\cdot), \boldsymbol{\eta}_l \sim \mathcal{GP}, \tag{22}$$

with covariance function,

$$\text{cov}\left(h_{l+1,i}(\mathbf{x}), h_{l+1,i}(\mathbf{x}') \mid h_{l,i}(\cdot), \boldsymbol{\eta}_l\right)$$
$$= \vartheta_l\left(h_{l,i}(\mathbf{x}), h_{l,i}(\mathbf{x}')\right) - \vartheta_l\left(h_{l,i}(\mathbf{x}), \boldsymbol{\beta}\right) \vartheta_l\left(\boldsymbol{\beta}, \boldsymbol{\beta}\right)^{-1} \vartheta_l\left(\boldsymbol{\beta}, h_{l,i}(\mathbf{x}')\right). \tag{23}$$

When $h_{l,i} \to h_{l+1,i}$ is a linear transformation as in (7), it induces a conditional GP prior:

$$h_{l+1,i}(\cdot) \mid h_{l-1,i}(\cdot) \sim \mathcal{GP}, \tag{24}$$

with covariance function,

$$\text{cov}(h_{l+1,i}(\mathbf{x}), h_{l+1,i}(\mathbf{x}') \mid h_{l-1,i}(\cdot))$$
$$= c_{l,i}(\mathbf{x}, \mathbf{X}_Q) k_{l,i}(\mathbf{X}_Q, \mathbf{X}_Q)^{-1} \text{cov}\left(\mathbf{h}_{l,i} \mid h_{l-1,i}(\cdot)\right) k_{l,i}(\mathbf{X}_Q, \mathbf{X}_Q)^{-1} c_{l,i}(\mathbf{X}_Q, \mathbf{x}'). \tag{25}$$

# F    Ablation Studies

We conducted a series of ablation studies to further evaluate our method.

**Various Hyperparameters.** We performed two comprehensive ablation studies — one on the real-world dataset *Beijing-Air* and the other on the simulated dataset *2D-Darcy* — to investigate

Table 3: DGPFM-FT ablations on *Beijing-Air*. The base model uses 4 integration layers, 10 Fourier modes, 256 channels, 32 inducing points, and a weighted Matern kernel (DOF 5/2 and 13/2). Best results are shown in bold.

(a) The number of integration layers.

| #Int Layers | 1 | 2 | 3 | 4 | 5 | 6 |
|---|---|---|---|---|---|---|
| NRMSE | 0.583 | 0.539 | 0.373 | 0.288 | 0.263 | **0.253** |
| NLL | N/A | 21.97 | 7.84 | **7.80** | 8.12 | 8.37 |

(b) The number of latent channels $C$.

| Channels ($C$) | 8 | 16 | 32 | 64 | 128 | 256 | 512 |
|---|---|---|---|---|---|---|---|
| NRMSE | 0.521 | 0.491 | 0.462 | 0.411 | 0.375 | 0.288 | **0.242** |
| NLL | 9.40 | 8.47 | 8.45 | 8.24 | 8.40 | **7.80** | 8.78 |

(c) The number of inducing points $S$.

| Inducing Points ($S$) | 4 | 8 | 16 | 32 | 64 | 128 |
|---|---|---|---|---|---|---|
| NRMSE | 0.271 | 0.311 | **0.269** | 0.288 | 0.270 | 0.273 |
| NLL | 20.04 | 8.99 | 8.73 | **7.80** | 8.54 | 9.09 |

(d) Number of Fourier modes.

| Modes | 4 | 8 | 12 | 16 | 32 | 64 |
|---|---|---|---|---|---|---|
| NRMSE | 0.391 | 0.305 | 0.277 | 0.255 | 0.224 | **0.219** |
| NLL | 7.94 | 8.38 | 7.91 | 8.13 | **7.81** | 8.30 |

(e) Choice of kernels.

| GP Kernel | Squared Exp | Matérn 5/2 | Matérn 13/2 | Weighted Matérn (5/2+13/2) |
|---|---|---|---|---|
| NRMSE | 0.321 | 0.289 | 0.308 | **0.288** |
| NLL | 8.26 | 8.26 | 8.21 | **7.80** |

the influence of hyperparameter choices. For *Beijing-Air*, we examined all major DGPFM-FT hyperparameters, including the number of integration (linear) layers, Fourier modes, latent channels ($C$), inducing points ($S$), and kernel/covariance choices. The base model consisted of 4 integration layers, 10 Fourier modes, 256 channels, 32 inducing points, and a weighted Matérn kernel (DOF 5/2 and 13/2). We varied each hyperparameter independently while fixing the others and evaluated performance using normalized root mean square error (NRMSE) and negative log-likelihood (NLL). To ensure statistical reliability, each configuration was run five times, and we report mean NRMSE and NLL values. The second study ablated DGPFM-QR hyperparameters on the Darcy Flow problem, with a base model of 5 integration layers, 64 latent channels, 64 inducing points, and the same kernel setup. Results are summarized in Table 3 and Table 4.

The ablations show that DGPFM-FT's expressivity improves with additional integration layers, channels, and Fourier modes, as reflected in NRMSE. However, beyond four integration layers or larger model sizes, NLL improvements diminish — likely due to the increasing difficulty of variational inference optimization. Varying the number of inducing points produced no clear trend, suggesting this hyperparameter should be tuned per dataset. Matérn kernels consistently outperformed the Squared Exponential, indicating that kernel smoothness has a significant impact on performance.

For DGPFM-QR, performance improved with more layers, channels, and inducing points, but NLL began to degrade once the number of inducing points exceeded 16. As with DGPFM-FT, finitely smooth kernels were advantageous; interestingly, the Matérn 5/2 kernel outperformed the 13/2 variant — the opposite of what we observed on *Beijing-Air*. This discrepancy, seen across datasets, motivated our use of weighted combinations of Matérn kernels to allow the model to adaptively learn optimal smoothness. As before, increasing model size improved NRMSE but yielded diminishing returns in NLL beyond a certain point.

Table 4: DGPFM-QR ablations on *Darcy-Flow*. The base model uses 5 integration layers, 64 channels, 64 inducing points, and a weighted Matern kernel (DOF 5/2 and 13/2). Best results are shown in bold.

(a) The number of integration layers.

| #Int Layers | 1 | 2 | 3 | 4 | 5 | 6 |
|---|---|---|---|---|---|---|
| NRMSE | 9.97e-2 | 2.67e-2 | 2.27e-2 | 1.97e-2 | 1.86e-2 | **1.80e-2** |
| NLL | N/A | 2.04 | -2.97 | -3.71 | -4.06 | **-4.13** |

(b) The number of latent channels $C$.

| Channels ($C$) | 8 | 16 | 32 | 64 | 128 | 256 |
|---|---|---|---|---|---|---|
| NRMSE | 2.47e-2 | 2.29e-2 | 1.99e-2 | 1.86e-2 | **1.72e-2** | 1.89e-2 |
| NLL | -2.46 | -3.64 | -3.94 | **-4.06** | -3.07 | -3.61 |

(c) The number of inducing points $S$.

| Inducing Points ($S$) | 4 | 8 | 16 | 32 | 64 | 128 |
|---|---|---|---|---|---|---|
| NRMSE | **1.82e-2** | 1.83e-2 | 1.87e-2 | 1.87e-2 | 1.86e-2 | 1.95e-2 |
| NLL | -3.34 | -3.97 | **-4.19** | -3.27 | -4.06 | -4.18 |

(d) Choice of kernels.

| GP Kernel | Squared Exp | Matérn 13/2 | Matérn 5/2 | Weighted Matérn (5/2+13/2) |
|---|---|---|---|---|
| NRMSE | 2.78e-2 | 1.92e-2 | 2.05e-2 | **1.86e-2** |
| NLL | -2.98 | -3.43 | -3.52 | **-4.06** |

**Projection Points.** Next, we examined the effect of the number of projection points, *i.e.,* the quadrature nodes or sampling locations used across GP layers. We performed an ablation study on the 1D Burgers' equation with varying numbers of projection points, using Gauss–Legendre quadrature. The quadrature resolution directly determines the dimensionality of the weight function and thus the number of trainable parameters.

As shown in Table 5, model performance degrades when the number of projection points is too small (*e.g.*, 8 or 16). However, beyond a certain threshold (64), additional points provide little to no improvement while substantially increasing the parameter count and computational cost.

Table 5: Performance of DGPFM-QR with Gauss-Legendre quadrature on 1D Burgers' equation.

| #Project Points | 8 | 16 | 32 | 64 | 128 | 256 |
|---|---|---|---|---|---|---|
| NRMSE (%) | 13.684 | 2.755 | 0.982 | **0.676** | 0.796 | 0.892 |

**GP Activation.** To assess the benefit of our GP-based activation, we conducted ablation studies comparing it against standard non-probabilistic activations commonly used in neural networks (ReLU and Tanh), as well as against the case with no nonlinear activation.

As shown in Table 6, the GP-based activation substantially improves both training and test errors relative to ReLU and Tanh (by more than 10%), with the sole exception of *Beijing-Air*, where the test error is marginally higher (a relative increase of 1.8%). Removing the nonlinear activation entirely results in a large increase in test error, underscoring its critical role in model performance.

Table 7: Performance of DGPFM-QR with different integral transforms on *2D Darcy*.

| | Dimension-wise | Full-dimensional |
|---|---|---|
| Training NRMSE (%) | 1.247 | **0.0830** |
| Test NRMSE (%) | **1.824** | 3.669 |

Table 6: DGPFM with different activate functions. The base models are the same in Table 3 and 4

(a) DGPFM-FT on *Beijing Air*.

| Activation | ReLU | Tanh | GP Activation | No Activation |
|---|---|---|---|---|
| Training NRMSE(%) | 0.6916 | 0.6953 | **0.028901** | 53.275 |
| Test NRMSE(%) | **26.363** | 37.402 | 26.854 | 57.615 |

(b) DGPFM-QR on *2D Darcy*.

| Activation | ReLU | Tanh | GP Activation | No Activation |
|---|---|---|---|---|
| Training NRMSE (%) | 1.581 | 1.737 | **1.247** | 5.395 |
| Test NRMSE (%) | 2.131 | 2.0292 | **1.824** | 6.703 |

**Dimension-Wise and Full Integral Transform.** To evaluate the effectiveness of our dimension-wise discrete integral transform, we trained a DGPFM-QR model on *2D-Darcy* with and without the dimension-wise transform, using five integration layers, 64 channels, 64 inducing points, and a weighted Matérn kernel (DOF 5/2 and 13/2). The trapezoidal rule with 29 projection points per input dimension was employed. As shown in Table 7, the full-dimensional transform fits the training data more closely but performs substantially worse on the test set, indicating clear overfitting.

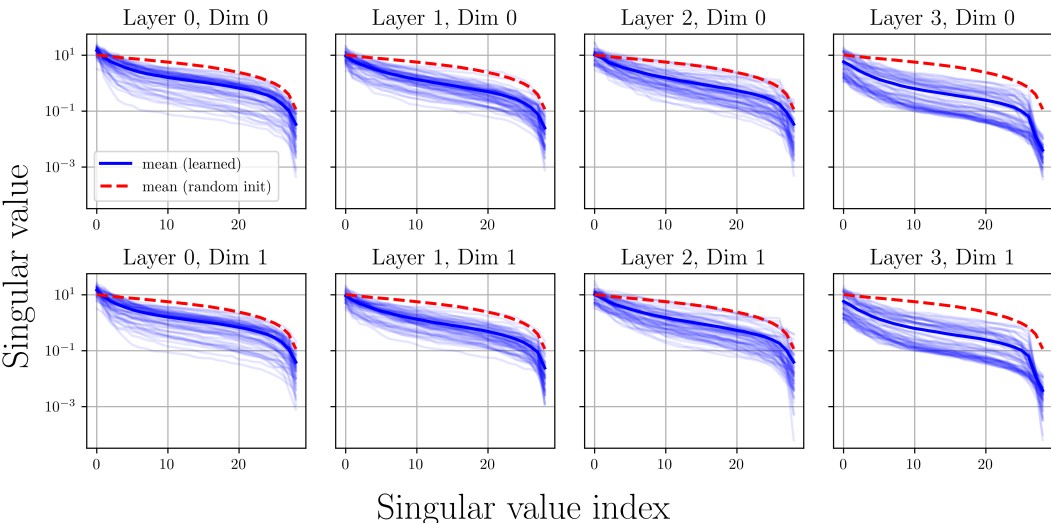

Figure 5: The Singular values of learned weight function values (matrices) versus randomly initialized matrices for running DGPFM-QR on *2D Darcy*.

# G  Visualization of Learned Weight Functions

To better understand the representations learned by the weight functions, we applied DGPFM-QR to the *2D Darcy* problem and analyzed the weight matrices in the integration (linear) layers. Specifically, we performed Singular Value Decomposition (SVD) on the weight matrix associated with each input dimension. As shown in Figure 5, in the early layers the singular values decay slowly, indicating that the weight matrices remain close to full rank and thus capture diverse, information-rich features. In contrast, in the layers closer to the output, the singular values decay more rapidly, suggesting the emergence of low-rank structures. This progression highlights a representational shift: early layers emphasize broad feature extraction, while later layers distill these features into more compact and structured representations tailored for prediction.

## H   Running Time

To evaluate training efficiency, we compared our method with the neural operator models GNOT and FNO on the *3D NS* and *2D Darcy* datasets. All experiments were conducted in the same computing environment: a Linux workstation equipped with an NVIDIA GeForce RTX 4080 GPU. We measured both the training time per epoch and the total training time (*i.e.,* until the stopping criterion was satisfied). The results are summarized in Table 8. We observe that DGPFM (including both DGPFM-FT and DGPFM-QR) and FNO achieve comparable performance in terms of both per-epoch time and overall training time. On *3D NS*, DGPFM (both variants) is faster than FNO, while on *2D Darcy*, DGPFM is slightly slower. In contrast, on both datasets, GNOT is around an order of magnitude slower than either DGPFM or FNO, likely due to the overhead introduced by its attention mechanism. Taken together, these results confirm the efficiency of our method: despite performing Bayesian learning, it remains nearly as efficient as neural network models that only conduct point estimation.

Table 8: Training time comparison. All the methods were run on a Linux workstation with a NVIDIA GeForce RTX 4080 GPU.

(a) *3D NS*

| Method | Per-epoch (seconds) / Step | Total (min) |
| --- | --- | --- |
| GNOT | 0.2213 | 3319.611 |
| FNO | 0.0383 | 570.375 |
| DGPFM-FT | 0.0218 | 326.947 |
| DGPFM-QR | 0.0221 | 331.990 |

(b) *2D Darcy*

| Method | Per-epoch (seconds) | Total (min) |
| --- | --- | --- |
| GNOT | 0.133 | 220.844 |
| FNO | 0.00945 | 15.750 |
| DGPFM-FT | 0.0241 | 40.239 |
| DGPFM-QR | 0.0184 | 30.622 |

## I   Limitation

Our current designs of the discrete integral transform are relatively simple and may be limited in scope. The associated weight functions are global over the domain and thus may struggle to capture local, nonstationary patterns. In future work, we aim to explore richer classes of transforms, such as spline-based or orthogonal basis functions, as well as localized sparse functions. These alternatives could further reduce model complexity, improve expressiveness, and provide finer control over the granularity of integration, thereby enabling the model to adapt more effectively to heterogeneous or highly localized structures in functional data.

