# OpenReview forum: "Deep Gaussian Processes for Functional Maps"
_ICLR.cc/2026/Conference — Submitted to ICLR 2026_

### Official Review · Reviewer_qnk8 · 2025-10-21

**Soundness:** 4
**Presentation:** 3
**Contribution:** 4
**Rating:** 8
**Confidence:** 3

**Summary:**

This paper proposes a deep Gaussian Process (GP) for function-to-function regression. I applaud the authors for working in this field. It's both exciting and highly relevant to push probabilistic ML forward. The paper is clearly very polished and well written. The experimental validation makes a strong case for the superiority of the proposed methodology. My comments mostly come in the form of questions, rather than requests for corrections.

**Strengths:**

- The research scope is highly relevant and impactful.
- Clean and concise writing.
- Clean nomenclature.
- Mathematical correctness.

**Weaknesses:**

- Some assumptions need a bit of explaining. I do not doubt that they are correct, but I believe that the reader might need some help understanding the reasoning in parts (see the questions).
- An introductory figure visualizing a core component is missing. I know that space is tight, but I think that an introductory figure could help less-familiar readers understand what the paper is trying to achieve. [OPTIONAL]
-There are just a couple of typos, for example, "sequence of linear and nonlinear transformation(s) as" and "To perform (a/the) nonlinear transform".

**Questions:**

"To flexibly accommodate varying sampling locations, and to enable tractable computation of the
GP layers, we introduce a set of fixed locations XQ to serve as the projection points for each GP
layer." This connection is unclear to me and needs clarification.   Why do flexibly varying sampling locations and tractable computation benefit from fixed locations?

 "To enrich representation, we then introduce a weight matrix." Again, unclear. How does a weight matrix enrich representation?

"Next, we apply a sequence of linear and nonlinear transformations as described in Section 3.1. To
simplify training and avoid the costly, complex computation of conditional covariance matrices, we
model the linear transformation using the GP conditional mean (interpolation) rather than the full
conditional Gaussian distribution." Why is this approximation an OK thing to do? I think it needs a defense.

---

### Official Review · Reviewer_24Ym · 2025-10-21

**Soundness:** 3
**Presentation:** 2
**Contribution:** 3
**Rating:** 4
**Confidence:** 3

**Summary:**

The authors propose a deep GP approach to learning functional mappings, allowing uncertainty quantification. Their key contributions include a tractable expression of the posterior, which is due to a discretization of the input space and a dimension-wise integral transform simplification. The authors show that their approach outperforms other baselines in several benchmarks.

**Strengths:**

The paper addresses an important problem with broad applications. Moreover, it considers uncertainty quantification, which is important.

Overall, the paper is mostly well written. The sentences are clear and easy to follow.

The method consists of a well-known framework (deep GPs), with some minor changes, making it not especially novel.

**Weaknesses:**

My main criticism is that the paper can be structured in a better way. Right now, most of the paper reads as a long derivation of the method, with little space for discussions and providing intuitive explanations. I also feel that the paper does not sufficiently address and discuss its limitations.

Though the paper is generally well-written, there are still some minor typos, and it would benefit from some proofreading. Articles, in particular, are frequently missing.

**Questions:**

Line 089: Is this really a projection? This is unclear to me.

Line 107: Colon instead of comma?

Line 108: Where is w introduced? It seems to me that it is implicitly defined, which is not clearly stated here.

Line 112: Is this similar to FLR or is it exactly FLR?

LIne 128-132: Could the authors provide a motivating example for the setup where Xin and Xout are not identical?

Line 134: what is a projection point?

Line 135: Is a component here a dimension? Is it reasonable to assume independence across dimensions?

Lines 133-147: This section is quite dense in notation. Since it’s just a derivation, I suggest moving it to the appendix, then presenting only the final model, along with a discussion and intuition about it.

Is there a drawback to using XQ as projection points?

Line 232: What are the potential additional errors? It would be useful to have a comparison between the full integral transform and the dimension-wise integral transform (even if just on a toy dataset) to estimate this.

Why does a variational learning algorithm “enable uncertainty quantification via probabilistic inference”? I thought the UQ aspect was already taken care of by the choice of deep GPs. Is this only to enable training?

The algorithm section does not detail an algorithm per se, but rather provides more derivation steps towards simplifying the model and obtaining a tractable training loss.

Will the authors make code available for reproducibility?

---

### Official Review · Reviewer_m5Av · 2025-10-23

**Soundness:** 3
**Presentation:** 2
**Contribution:** 2
**Rating:** 4
**Confidence:** 4

**Summary:**

This paper proposes DGPFM, applying deep Gaussian processes to function-on-function regression. The method constructs mappings through sequential GP-based linear integral transforms and nonlinear GP activations. The authors leverage discrete approximations to simplify covariance computations and develop variational inference with inducing points. Experiments on PDE benchmarks and real-world datasets show competitive predictive performance and uncertainty quantification.

**Strengths:**

1. The method accommodates sparse, noisy, and irregularly sampled functional data, which is valuable for real applications.

2 The observation that discrete approximations cause covariance cancellation (Eq. 11) provides implementation simplification.

3 Testing across 6 datasets (3 PDE simulations, 3 real-world) with multiple baselines.

4 Provides probabilistic predictions, unlike deterministic neural operators.

**Weaknesses:**

1.The paper fails to adequately review the extensive Deep GP literature. There is no discussion of recent advances in deep GP architectures and inference. The paper also misses important connections to functional data analysis with GPs, operator-valued kernels, and multi-output GPs. It reads as if DGPs were applied to functional maps for the first time, but the connection is quite straightforward given existing DGP and functional GP literature. The Related Work section (Section 5) only briefly mentions Salimbeni & Deisenroth (2017) but ignores the extensive follow-up work on deep GPs with structure, recent advances in inducing point selection, and the rich literature on operator-valued kernels for functional data.

2.The paper doesn't convincingly argue why deep GP layers are necessary versus simpler alternatives such as single-layer GP with expressive kernels (spectral mixture, deep kernels), GP with neural network mean functions, or simpler probabilistic models like Bayesian neural networks. Table 1 evidence is weak. On Beijing-Air, FLR-GP (single GP layer) achieves 0.552 NRMSE while DGPFM-QR gets 0.263, but GNOT (deterministic) gets 0.553. The gain over single GP is not clearly due to depth. On 3D-NS, FLR methods outperform all deep methods including DGPFM, directly contradicting the depth hypothesis. The ablation in Table 3a shows improvement from 1 to 4 layers, but diminishing returns after layer 4, and NLL actually degrades, suggesting overfitting rather than fundamental benefits of depth.
The connection to neural operators is superficial. Neural operators use Fourier/kernel integral transforms and so does DGPFM, but the paper doesn't explain what GPs add beyond uncertainty quantification that could be achieved via ensembles.

3.Table 1 shows modest or inconsistent gains. On 1D Burgers, DGPFM-FT achieves 1.79e-3 versus FNO's 2.76e-3, an improvement but FNO is much simpler. On 2D Darcy, DGPFM-FT gets 1.67e-2 versus FNO's 1.79e-2, only a marginal 7% improvement. On 3D NS, DGPFM loses to simple FLR-Fourier (4.51e-1 vs 4.67e-1). On Beijing-Air, DGPFM-QR (0.263) beats GNOT (0.553), but GNOT is not well-tuned for irregularly sampled data, making this an unfair comparison. On Quasar, DGPFM-QR (0.0045) versus GNOT (0.0054) shows 20% improvement, but at what computational cost?
Regarding uncertainty quantification, while MNLL is better, the comparison against MC-Dropout and SGLD may not be fair. SGLD hyperparameters (learning rate from {1e-6,...,1e-2}) seem poorly tuned, MC-Dropout rates {0.1,...,0.5} are limited, and there is no comparison with deep ensembles, which often provide well-calibrated uncertainty with less cost. The improvements don't justify the added complexity of deep GPs with multiple approximations (inducing points, dimension-wise transforms, Dirac delta linear layers).

4.The ablation studies in Appendix F have critical gaps. There is no ablation on DGP-specific components such as the contribution of GP activations versus deterministic activations (Table 6 shows this partially, but results are mixed), the benefit of inducing points versus full GP, or the impact of whitening transformation claimed as "critical" but not ablated.
Table 6 is concerning. GP activation is worse than ReLU on Beijing-Air test set (26.854 vs 26.363), and only marginally better than Tanh on 2D Darcy (1.824 vs 2.029). This undermines the core motivation for using GPs. The paper also lacks comparisons with single deep GP layer plus more sophisticated kernels, or deterministic integral transforms plus uncertainty via ensembles.

5.The dimension-wise approximation in Eq. 12 is a strong approximation with no theoretical justification. When is w(x₁,x₂,y₁,y₂) ≈ w₁(x₁,y₁) + w₂(x₂,y₂) valid? Table 7 shows dimension-wise transform has worse training fit (1.247% vs 0.083%) but better test (1.824% vs 3.669%), suggesting the full model is overfitting, not that dimension-wise is a good approximation. This approximation is crucial for scalability but breaks the GP interpretation.
Using Dirac delta in Eq. 7 means linear transforms are deterministic, not probabilistic. This contradicts the GP framework. Why not use full conditional distributions? The model mixes probabilistic (GP activations) and deterministic (linear transforms) components without clear justification.
The model involves an approximation cascade: GP conditional mean (not full distribution), dimension-wise transforms, inducing points, and discrete quadrature. Each introduces error, but the cumulative effect is not analyzed.

6.O(dN²_Q BL) scales poorly with input dimension d. Table 8 shows comparable training time to FNO, but only for d≤3, and doesn't account for hyperparameter search cost (extensive, Appendix C) or inference time. Inducing points S from {32,...,512} require expensive tuning. Memory cost of storing multiple GP layers is not discussed.

7. No code is provided and many implementation details are missing.

**Questions:**

1.What specifically motivated using deep GPs versus single-layer GPs with expressive kernels? The ablations show diminishing returns.

2,How do you justify the dimension-wise approximation theoretically? When does it fail?

3,Why does DGPFM fail on 3D NS where simple FLR succeeds?

4,Can you compare against simpler probabilistic baselines such as deep ensembles of neural operators or Bayesian NNs with modern VI?

5.What is the contribution of each approximation (GP conditional mean, dimension-wise, inducing points)? An ablation removing them one-by-one would be valuable.

6.How does the method scale to d>5 and N>10,000?

---

### Official Review · Reviewer_y8ot · 2025-10-29

**Soundness:** 2
**Presentation:** 2
**Contribution:** 2
**Rating:** 4
**Confidence:** 4

**Summary:**

The authors propose a deep GP architecture for function-on-function regression and operator learning. The method interleaves full-fledged GP layers, linear transformations, and GP-based interpolation (kernel ridge regression). Their inference procedure leverages inducing points for the GPs modelling the non-linear transformations. The authors also propose a clever choice of quadrature points to alleviate the computational burden of numerical integration in the linear layers.

**Strengths:**

* The paper addresses a significant problem in operator learning: combining expressivity with calibrated uncertainty under noisy and irregularly sampled observations.

* While few, the empirical results show competitive or superior prediction performance compared to relevant benchmarks.

* The implementation insights for simplifying integral transform computations are practical and potentially broadly useful.

**Weaknesses:**

* **Clarity of modeling and notation**: The exposition would benefit from a clearer separation between the model definition and the inference procedure. The same mathematical form is reused for qualitatively different transformations (e.g., Eqs. 1 and 4), which can make the narrative difficult to follow. Annotating linear vs. nonlinear layers explicitly (e.g., ( \ell+1 \in \Gamma_{\text{lin}} ) vs. ( \Gamma_{\text{non}} )) could improve readability.

* **Novelty is somewhat overstated**: There are prior works on uncertainty-aware operator learning (e.g., [1,2]), and the contributions over these are not fully articulated. The positioning within the existing literature could be more precise.

* **Scalability not sufficiently demonstrated**: The paper proposes strategies to reduce quadrature costs, but there is no direct empirical validation of runtime or memory improvements. All experiments are low-dimensional, so practical scalability remains unclear.

* **Limited experimental coverage**: Despite citing PDEBench, only three PDE datasets are used. The evaluation does not fully test the method’s applicability to a broader range of PDE problems or higher-dimensional settings.

* **Missing efficiency comparisons**: The empirical section lacks wall-clock comparisons against strong operator learning baselines (e.g., FNO-based methods), making it difficult to assess efficiency trade-offs.

**References**

[1] *A Bayesian Neural Operator for Learning Parametric PDEs*. OpenReview: [https://openreview.net/pdf?id=6WvIkYsMA8](https://openreview.net/pdf?id=6WvIkYsMA8)

[2] *Infinite Neural Operators: Gaussian Processes on Functions*. arXiv: [https://arxiv.org/abs/2510.16675](https://arxiv.org/abs/2510.16675)

**Questions:**

* If I understand correctly, the linear transformations are deterministic mappings (because of the interpolation trick). Meanwhile, the non-linear transformations are sampled from a GP. Am I correct?

* Equation for p(joint) might be confusing if p(H_{l+1}|H_{l}) is a dirac delta for all l+1 in \Gamma_{lin}, right?

* Computational complexity: I guess this is the complexity per epoch?

* Experiments in very low dimensions (1 to 3). How well does this method scale with dimensions? Can we measure the usefulness of the integration tricks?

* Can you provide a time comparison between your method and the baselines from Table 1?

* The hypothesis you raised that NS favors linear models with simple basis functions (due to few samples) can be tested by simply increasing the dataset size.

---

### Meta-Review · Area_Chair_PBVo · 2026-01-02

**Summary:**

This paper introduces a deep Gaussian process (DGP) approach to function-on-function regression, motivated by a desire for non-linear function mapping, applicability in the data-sparse regime, and uncertainty quantification. Though the idea is well motivated and could provide utility within the operator learning subfield, the paper is unsuitable for publication in its current form. Notable weaknesses include:

1. A lack of algorithmic/implementation details. The authors provide a high-level set of steps for deriving their method, but do not include the specific details needed to reproduce their experiments or the method itself.
2. Lack of evidence for why deep GPs are necessary rather than single-layer GPs with expressive kernels. The experimental evidence is mixed, not showing significant advantages from using the proposed method or using a deeper (i.e., more complex) transformation.
3. A lack of ablation studies, which are crucial for the proposed framework that relies heavily on approximations (discretization of the integral, variational inference, inducing points, etc.).
4. Limitation to low-dimensional problems. Scalability is poor due to the reliance on quadrature approximations, and the authors do not offer evaluations of strategies to reduce quadrature costs.

The authors did not provide a rebuttal.

**Reviewer Concerns:**

N/A (no rebuttal)

**Reviewer Scores:**

N/A (no rebuttal)

---

### Decision · Program_Chairs · 2026-01-26

Reject